# Species Diversity, Biomass and Carbon Stock Assessment of Kanhlyashay Natural Mangrove Forest

**Wai Nyein Aye** [1] , **Xiaojuan Tong** [2,*] **and Aung Wunna Tun** [2]

1   School of Forestry, Beijing Forestry University, Tsinghua East Road, Haidian District, Beijing 100083, China; wainyeinaye@gmail.com
2   School of Ecology and Nature Conservation, Beijing Forestry University, Tsinghua East Road, Haidian District, Beijing 100083, China; aungwunnnatunwunna@gmail.com
*   Correspondence: tongxj@bjfu.edu.cn

**Abstract:** Mangrove ecosystems sequester and store large amounts of carbon in both biomass and soil. In this study, species diversity, the above and below-ground biomass as well as carbon stock by the mangroves in Kanhlyashay natural mangrove forest were estimated. Six true mangrove species from four families were recorded in the sample plots of the study area. Among them, *Avicennia officinalis* L. from the Acanthaceae family was the abundance of species with an importance value of 218.69%. Shannon–Wiener's diversity index value (H′ = 0.71) of the mangrove community was very low compared to other natural mangrove forests since the mangrove stands in the study site possessed a low number of mangrove species and were dominated by a few species. Estimated mean biomass was $335.55 \pm 181.41$ Mg ha$^{-1}$ (AGB = $241.37 \pm 132.73$ Mg ha$^{-1}$, BGB = $94.17 \pm 48.73$ Mg ha$^{-1}$). The mean overall C-stock of the mangrove stand was $150.25 \pm 81.35$ Mg C ha$^{-1}$ and is equivalent to $551.10 \pm 298.64$ Mg $CO_2$ eq. The role of forests in climate change is two-fold as a cause and a solution for greenhouse gas emissions. The result of the study demonstrated that the mangroves in Letkhutkon village have high carbon storage potential, therefore it is necessary to be sustainably managed to maintain and increase carbon storage. Climate change mitigation may be achieved not only by reducing the carbon emission levels but also by maintaining the mangrove ecosystem services as carbon sinks and sequestration.

**Keywords:** biomass; carbon stock; allometric models; natural mangrove forest; Myanmar

## 1. Introduction

Mangrove forests are one of the most productive and diverse ecosystems in the world and provide significant ecological, economic, and social benefits [1]. The ecological benefits supported by the mangrove forests are bio-protection from littoral erosion [2,3], shoreline stabilizations, reducing the devastating impact of hurricanes, waves, and tsunamis, and protection from cyclones [3,4]. Additionally, mangrove ecosystems have a high carbon sequestration capacity, which is reflected in high aboveground biomass, high net primary production (NPP), the low decomposition rate of mangrove sediments, and belowground to aboveground biomass rations [5–8]. Mangrove forests have a critical role in climate change mitigation because they are able to absorb and store 3–5 times more carbon than other upland forests, mainly in soil [5]. Despite accounting <1% of the world's tropical forest area [9], mangroves account for 3–4% of global carbon sequestration by the total tropical forest area [10,11] and contribute 10–15% to the carbon sequestered by the world's ocean [9]. Globally, the average carbon stock of the mangrove ecosystem is 1023 Mg ha$^{-1}$ [5]; consequently, mangrove ecosystems are now being recognized for their pivotal role in global climate change mitigation. Concerning the characteristics of high carbon reservation and huge ecological benefits, mangrove ecosystems are eligible for inclusion in the United Nation's Reduce Emissions from Deforestation and Forest Degradation and to Enhance Carbon Stocks (REDD$^+$)

strategies [12] as well as the payments for ecosystem services (PES) [13] initiatives that are emerging in many countries. On the other side, the deforestation rate of mangrove forests is still higher than inland terrestrial forests. Globally, it is estimated that mangrove forests have been lost with an annual average rate of 0.16% to 0.39% [14] and pose a significant risk to carbon emission as a consequence of mangrove deforestation.

Mangrove forests in Myanmar grow along the 2832 km-long coastlines, oriented along the Bay of Bengal and the Andaman Sea [15]. The total area of mangroves in Myanmar reaches 3.3% of the total area of mangroves of the world [15,16], and mangrove forest types can be divided into Delta Mangrove and Coastal Mangrove [17]. Mangrove ecosystems provide important services such as ecological, economic, and environmental benefits to local people; however, mangrove coverage in Myanmar has decreased by more than half of the total mangrove area over the past three decades. Myanmar is regarded as the current mangrove deforestation hotspot globally [18] with the highest annual rates (~1%) of mangrove deforestation and third-highest potential annual $CO_2$ emissions (784 kg $CO_2$eyr$^{-1}$) [19]. The biggest drivers of mangrove deforestation in Myanmar are over-exploitation, illegal felling, agricultural expansion, and conversion to fish and shrimp ponds [20]. The inventory of carbon stocks in mangrove ecosystems is limited, and only a few studies have quantified the carbon stocks of these ecosystems in Myanmar.

Forest biomass is regarded as an important variable in quantifying the role of forests in the carbon cycle [21]; thus, the estimation of biomass is crucial for studying the carbon cycle of the forest ecosystem. Allometric models are widely applied for biomass estimation of mangrove forests [22] and allometric equations for biomass estimation are developed by applying physical parameters of the tree, such as height, diameter at breast height, basal area, density, and their combination. The objectives of the present study were to (i) estimate the species diversity of mangrove stands, (ii) evaluate the potential of biomass and carbon stock, and (iii) explore the relationships of stand-level carbon stock to stand structural variables such as mean diameter, mean height, basal area and their combination.

## 2. Materials and Methods

### 2.1. Description of Study Site

The research was carried out in the Kanhlyashay natural mangrove forests located at the estimated coordinates 16°21′26.93″ N and 96°13′02.37″ E. The mangrove stands naturally reappeared on the mudflat after the Cyclone Nargis wreaked havoc in 2008. The abundant growth of mangroves lies around 1–8 m above the current level of the sea and covers approximately 197 ha (487 acres) along the banks of the sea; then, the mangroves serve as a natural barrier against natural disasters such as sea-level rise, storm surges, and floods and help minimize the damage done to property and life in the Letkhutkon village located in the Kungyangon township of Yangon Division. The Kanhlyashay natural mangrove forest has planned to designate as a protected public forest by the Forest Department, Ministry of Natural Resources and Environmental Conservation, Myanmar. The soil formation in the study site has been underlain by the intertidal mudflats that are essential habitats for many fishes as nurseries and feeding grounds. Most of the local communities in the Letkhutkon village depend on small-scale marine fisheries for their livelihood; this area has an average precipitation of 2375 mm, with an average temperature of 26.8 °C and a tropical monsoon climate [23]. The study site was selected based on accessibility and safety in going to and from the natural mangrove stands. Additionally, this study is the first comprehensive forest inventory in the Kanhlyashay natural mangrove forests. The natural mangrove formation has been dominated by *Avicennia officinalis* L. in association with *Sonneratia apetala* Buch. Ham., *S. caseolaris* (L.) Engl., and *Aegiceras corniculatum* (L.) Blanco; then, *Avicennia alba* Blume and *Bruguiera sexangula* (Lour.) Poir. are rarely observed in the mangrove stand.

The fieldwork was performed from June to July 2021 during the rainy season. Letkhokkon village near the Andaman Sea has equal lengths of dry and rainy seasons. The wet season is oppressive and overcast, the dry season is muggy and partly cloudy, and it is hot year-

round. Throughout the year, the temperature typically varies from 19 °C to 36 °C and is rarely below 17 °C or above 39 °C. The hottest month of the year is April with an average maximum temperature of 37 °C and the coldest month is January with an average low of 19 °C and a high of 32 °C. A map of the research stations was presented in Figure 1.

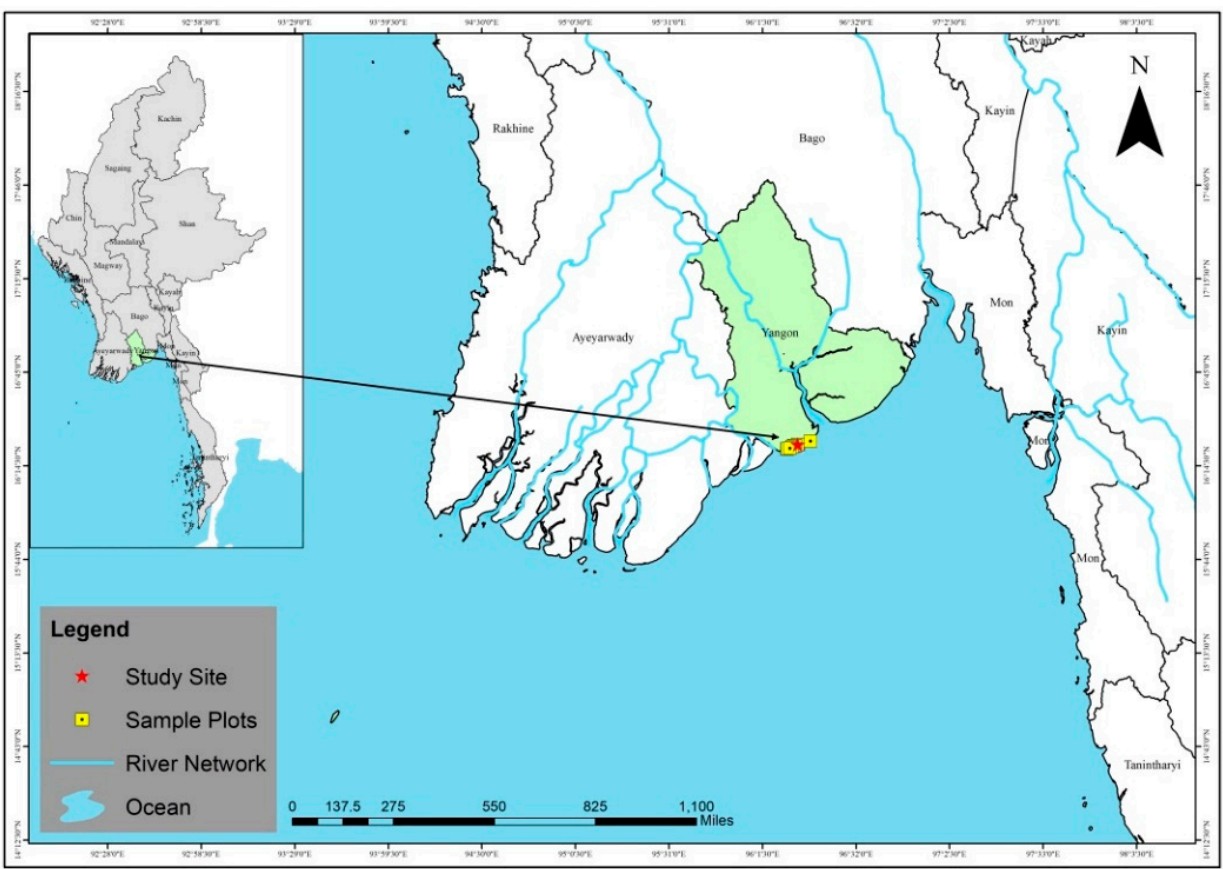

**Figure 1.** Location of the study site and sampling points in the mangrove stand of Kanhlyashay natural mangrove forest.

*2.2. Data Collection*

A total of 25 sampling plots of 400 m$^2$ were established through a non-destructive quadrat sampling technique to determine the species composition, biomass, and carbon stock in the study area. The total sampling area covered was 0.5% (1 ha) of the total area. A global positioning system (GPS) was used to mark the spatial location of each sampling pot. Within each sampling plot, all trees with a diameter at breast height (dbh) of ≥5 cm were measured, identified, and counted. A diameter tape was used to measure the dbh; total tree heights were estimated using Suunto Clinometer. The dbh of Bruguiera and Rhizophora species were determined by measuring the trunk diameter at 30 cm above the buttress and above the highest prop root, respectively, whereas the dbh of the rest was measured at 130 cm above ground [24]. The distribution of stand density, species composition, biomass, and carbon stock per plot of the natural mangrove stand in LetKhutKon Village was described in Table 1.

**Table 1.** Distribution of stand density, biomass, carbon stock, and $CO_2$ equivalent in the natural mangrove stand, LetKhutKon Village.

| Plot | Stand Density (Stems ha$^{-1}$) | Species | DBH Range (cm) | Height Range (m) | Basal Area (m$^{2.5}$ ha$^{-1}$) | Biomass (Mg ha$^{-1}$) | | | C-Stock Mg C ha$^{-1}$ | $CO_2$ Equivalent (MgCO$_2$ eq) |
|---|---|---|---|---|---|---|---|---|---|---|
| | | | | | | AGB | BGB | TB | | |
| 1 | 1250 | Ao, Sa | 5.00–29.00 | 3.05–9.75 | 19.950 | 131.077 | 55.307 | 186.38 | 83.18 | 305.26 |
| 2 | 1125 | Ao, Sa | 8.00–34.00 | 3.05–8.23 | 21.725 | 145.769 | 61.048 | 206.82 | 92.32 | 338.82 |
| 3 | 875 | Ao, Sa | 5.50–34.00 | 2.44–7.93 | 26.000 | 214.807 | 83.486 | 298.29 | 135.52 | 490.01 |
| 4 | 875 | Ao | 5.00–35.30 | 2.13–8.53 | 19.375 | 159.381 | 62.136 | 221.52 | 99.14 | 363.14 |
| 5 | 875 | Ao | 5.20–31.80 | 2.74–8.23 | 25.725 | 211.055 | 82.610 | 293.67 | 131.41 | 482.29 |
| 6 | 875 | Ao | 5.30–37.00 | 2.74–8.23 | 33.900 | 294.506 | 111.881 | 406.39 | 182.05 | 668.13 |
| 7 | 625 | Ao | 16.40–40.00 | 5.18–9.45 | 42.950 | 405.162 | 147.636 | 552.80 | 248.00 | 910.18 |
| 8 | 1000 | Ao | 5.50–40.70 | 2.74–9.14 | 47.750 | 425.534 | 159.583 | 585.12 | 262.24 | 962.42 |
| 9 | 1625 | Ao | 7.60–41.20 | 3.05–9.14 | 69.125 | 603.928 | 228.753 | 832.68 | 373.06 | 1369.13 |
| 10 | 1000 | Ao | 6.60–37.20 | 4.27–9.50 | 30.025 | 248.525 | 96.739 | 345.26 | 154.54 | 567.14 |
| 11 | 1125 | Ao | 6.00–40.10 | 3.66–9.45 | 41.025 | 352.219 | 134.529 | 486.75 | 218.01 | 800.09 |
| 12 | 750 | Ao | 6.10–39.50 | 3.35–9.14 | 29.575 | 259.835 | 98.089 | 357.92 | 160.38 | 588.58 |
| 13 | 1375 | Ao | 5.00–41.30 | 3.05–9.14 | 51.650 | 446.873 | 170.086 | 616.96 | 276.36 | 1014.26 |
| 14 | 1525 | Ac, Ao, Sa | 5.00–26.60 | 2.13–8.23 | 17.075 | 113.658 | 49.255 | 162.91 | 72.63 | 266.55 |
| 15 | 1075 | Ao, Sa | 5.90–28.80 | 2.13–9.14 | 24.325 | 187.260 | 75.120 | 262.38 | 117.31 | 430.52 |
| 16 | 1875 | Ao, Sa, Sc | 5.10–35.60 | 2.13–9.50 | 41.225 | 302.509 | 121.188 | 423.70 | 189.44 | 695.26 |
| 17 | 975 | Ao | 5.50–38.80 | 2.13–8.84 | 30.275 | 258.542 | 99.003 | 357.55 | 160.13 | 587.66 |
| 18 | 2025 | Ao | 5.40–37.60 | 2.13–9.14 | 43.800 | 346.801 | 138.028 | 484.83 | 216.83 | 795.76 |
| 19 | 1075 | Ao | 6.10–29.00 | 3.66–8.84 | 22.900 | 171.747 | 70.439 | 242.19 | 108.19 | 397.07 |
| 20 | 925 | Ao | 8.30–23.50 | 3.35–8.23 | 17.750 | 126.148 | 53.263 | 179.41 | 80.06 | 293.83 |
| 21 | 1175 | Ao, Sa, Sc | 6.50–37.00 | 2.13–7.62 | 30.025 | 229.082 | 91.594 | 320.68 | 143.39 | 526.24 |
| 22 | 1075 | Ac, Aa, Ao, Bs, Sc | 5.10–36.70 | 1.70–7.93 | 22.875 | 166.665 | 66.580 | 233.25 | 104.30 | 382.78 |
| 23 | 950 | Ac, Ao, Sa | 5.00–31.00 | 1.50–6.86 | 17.200 | 122.801 | 49.746 | 172.55 | 77.12 | 283.02 |
| 24 | 550 | Ac, Sa | 5.00–17.00 | 2.13–5.49 | 4.925 | 26.691 | 12.363 | 39.05 | 17.37 | 63.73 |
| 25 | 950 | Ac, Sa, Sc | 5.50–25.90 | 2.44–7.62 | 14.800 | 83.713 | 35.868 | 119.58 | 53.33 | 195.73 |
| | Mean | | 6.22–33.94 | 2.76–8.53 | 29.838 | 241.372 | 94.173 | 335.55 | 150.25 | 551.10 |
| | Standard deviation | | 2.32–6.26 | 0.84–0.96 | 14.032 | 132.731 | 48.728 | 181.41 | 81.35 | 298.64 |

Note: Ao-Avicennia officinalis; So-Sonneratia caseolaris; Sa-Sonneratia apetala; Ac-Aegiceras corniculatum; Bs-Bruguiera sexangula; Aa-Avicennia alba.

*2.3. Species Composition and Diversity*

Species composition and diversity were calculated based on the forest inventory data. Species composition is the number of different species in the study area; it can be described in terms of relative density (RD), relative frequency (RF), and relative basal area (RBA). The importance value index (IVI) provides an overview of the influence or role of a type of mangrove species in the community. Importance values of a species range from 0–300%, and tree species having an IVI of more than 10% were considered dominant tree species in this study. The formula used to calculate RD, RBA, RF, and IVI were listed below.

$$\text{RD = (Number of individuals of a species/Total number of individuals of all species)} \times 100 \tag{1}$$

$$\text{RBA = (combined BA of a species/total BA of all species)} \times 100 \tag{2}$$

$$\text{RF = (frequency of a species/sum of all frequencies)} \times 100 \tag{3}$$

$$\text{IVI = RD + RBA + RF} \tag{4}$$

The basal area was calculated as

$$\text{BA/Tree (m}^2) = \frac{\pi \times (\text{DBH})^2 \times 0.0001}{4} \tag{5}$$

where $\pi$ = a constant (3.146); DBH = diameter at breast height (cm), 0.0001 is a constant used to convert the measured centimetre square into meter square.

$$\text{Total Stand Basal Area (m}^2/\text{ha)} = \frac{\text{Sum of basal area for each tree}}{0.04} = \text{Sum of basal area} \times 25 \tag{6}$$

where 0.04 is plot size in hectare and 25 is a constant used to extrapolate the measurement of the basal area from per plot (m$^2$/plot) to per hectare (m$^2$/ha).

The Species Diversity index, determined in this study using the Shannon–Wiener's Index [25], indicated a quantitative description of mangrove habitat in terms of species distribution and evenness; this species diversity index was used in several studies [26–28] and was calculated using the following form:

$$H' = -\sum P_i \ln P_i \tag{7}$$

where:

$H'$ = the value of the Shannon–Wiener diversity index
$P_i$ = the proportion of ith species individuals to total species individuals
$\ln$ = the natural logarithm of $P_i$

$$\text{Evenness Index, E} = H'/\ln(S) \tag{8}$$

where, S = Number of species in the study area

Aboveground and Belowground Biomass Estimation and Carbon Stocks

Inside each plot, all mangrove trees $\geq$ 5 cm in diameter were identified according to W. Giesen et al. [29] and measured the trunk diameters (cm) and total height (m) for estimating above and below-ground biomass. Tree measurements, including diameter at breast height (dbh) and height (H) in sample plots, were converted into tree biomass by using an allometric equation (tree biomass equation) and then into carbon stock. Here, allometric equations adapted from Komiyama et al. [30] were used to estimate AGB and BGB as shown in (Equations (9) and (10)). The reason for choosing these allometric equations was they utilized mangroves of Southeast Asia as samples when developing the equations, and were favoured by many researchers as they didn't require tree height data.

The mean value of wood density ($\rho$) of each species was obtained from the Global Wood Density Database [31] by using the *getWoodDensity* function from the "BIOMASS" package

in R program. Then, total aboveground and belowground biomass production in the plots were obtained by summing the biomass of all the standing trees and the biomass of each sample plot had been converted to stand-level biomass (Mg ha$^{-1}$). Then, carbon stock of aboveground and belowground biomass showed in mega-grams per hectare (Mg C ha$^{-1}$).

$$AGB = 0.251\rho\ D^{2.46} \tag{9}$$

$$BGB = 0.199\ \rho^{0.899}\ D^{2.22} \tag{10}$$

where:

AGB (kg) = aboveground biomass estimates in kg per tree
BGB (kg) = belowground biomass estimate in kg per tree
D = diameter at breast height (dbh) in cm
$\rho$ = wood density in g cm$^{-3}$

The AGB and BGB were converted to above and below-ground carbon stock by multiplying 0.47 and 0.39 as a conversion factor [32–36] using the equations below:

$$\text{Aboveground carbon stock} = AGB \times 0.47 \tag{11}$$

$$\text{Belowground carbon stock} = BGB \times 0.39 \tag{12}$$

### 2.4. Statistical Analyses and Modelling Work

Regression analysis was used to establish allometric relationships of stand-level aboveground biomass carbon stock (Mg C ha$^{-1}$) with mean DBH (cm), mean height (m), and stand basal area (m$^2$ ha$^{-1}$). In forest biomass studies, the error variances for the allometric non-linear equations based on arithmetical units of measurement were not constant over all observations (heteroscedasticity) in most cases [37,38]. Using log-transformed data for linear regressions was the most commonly used method for the estimation of parameters in non-linear models to eliminate the effects of heteroscedasticity [38]. To minimize the systematic bias, a correction factor (CF) was calculated for each model [39]. The stand-level aboveground carbon stock models based on structural variables such as mean DBH, mean H, and BA can be expressed as follows:

$$\text{Model 1}: \ \ln(C) = \ln\ a + b\ \ln(\overline{D}) + \varepsilon \tag{13}$$

$$\text{Model 2}: \ \ln(C) = \ln\ a + b\ \ln(\overline{H}) + \varepsilon \tag{14}$$

$$\text{Model 3}: \ \ln(C) = \ln\ a + b\ \ln(BA) + \varepsilon \tag{15}$$

$$\text{Model 4}: \ \ln(C) = \ln\ a + b\ \ln(BA) + c\ \ln(\overline{H}) + \varepsilon \tag{16}$$

where, C = above-ground carbon stock (Mg ha$^{-1}$), BA = basal area (m$^2$ ha$^{-1}$), $\overline{D}$ = mean DBH, $\overline{H}$ = mean height (m), a, b and c = regression coefficients.

All the statistical analyses were performed using R programming software version-R 4.1.1. Before statistical analyses, data were checked to meet the requirements of normal distribution and variance homogeneity. All variables were tested for normality using a Shapiro–Wilk test. Logarithmic transformation was applied to both dependent and independent variables when the statistical requirements were violated. Data were analyzed through linear regression models. Pearson correlation test was used to analyze the relationship among structural characteristics. Finally, equation performance was carried out using various goodness-of-fit statistics, namely the adjusted coefficient of determination (R$^2$-adj), root mean squared error (RMSE) value, Akaike information criterion (AIC), Bayesian Information Criteria (BIC), Breusch–Pagan Test (bptest), Durbin–Watson test and *p*-value.

$$R^2\ adj = 1 - \frac{(n-1)\sum_{i=1}^{n}(yi - \hat{y}i)^2}{(n-p)\sum_{i=1}^{n}(yi - \overline{y}i)^2} \tag{17}$$

$$AIC= -2\text{logLik} + 2(p + 1) \tag{18}$$

$$BIC = -2\text{logLik} + (p + 1) \log (n) \tag{19}$$

$$RMSE = \sqrt{\frac{\sum_{i=1}^{n}(yi - \hat{y}i)^2}{n - p}} \tag{20}$$

where yi = observed value, ŷi = the estimated value, ȳi = the mean value of the observed carbon stock; n = the number of samples; p = the number of parameters, and logLik = the log-likelihood values of the non-linear regression model.

## 3. Results and Discussion

### 3.1. Species Composition

The mangrove stand in the study area comprised six true mangrove species, namely: *Avicennia officinalis* L., *A. alba* Blume, *Sonneratia apetala* Buch.-Ham., *S. caseolaris* (L.) Engl., *Aegiceras corniculutum* (L.) Blanco and *Bruguiera sexangula* (Lour.) Poir., belonging to four families. Mangrove species are classified as true mangrove or associated mangrove based on the criteria of Tomlinson [40]. True mangrove species of *Nypa fruticans* Wurmb, and a few associated mangrove species such as *Derris trifoliate*, *Imperata cylindrica* (L.) P.Beauv., were also found, but were not considered in biomass calculations. Mangrove species recorded at the study site were among the 44 true mangrove species thriving in Myanmar [41]. A total of 1102 individuals were enumerated from the 25 (20 × 20) plots. Among them, 78.77% were found to be of a single species, *A. officinalis* belonging to Acanthaceae family. *S. apetala* and *S. caseolaris* from the Lythraceae family and *Aegiceras corniculutum* from the Myrsinaceae family were the other major species occupying 13.97%, 4.17%, and 2.90% of the total species recorded from the study site. The remaining 0.09% was collectively represented by *A. alba* from the Acanthaceae family and *B. sexangula* from the Rhizophoraceae family. Most of these two species have dbh < 5 cm, which was below the threshold for biomass determination using the allometric equations. Figure 2 explained the species distribution of mangroves in LetKhutKon Village.

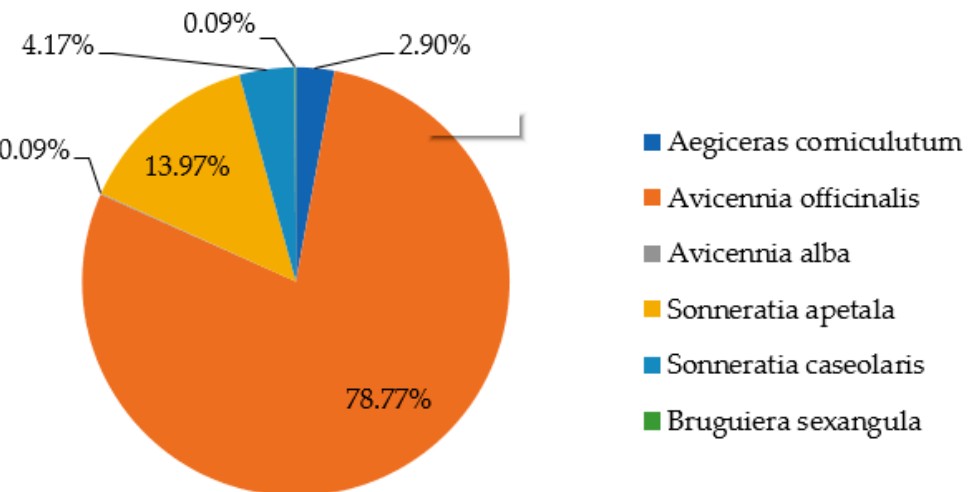

**Figure 2.** Species Distribution of Mangrove stands in LetKhutKon Village.

Densities of mangroves in the 1-ha sample area ranged from 550 to 2025 trees per ha (mean 1102 ± 353 stems ha$^{-1}$); a total basal area of the stand was 745.95 m$^2$ ha$^{-1}$ (mean ± sd = 29.84 ± 14.03 m$^2$ ha$^{-1}$) and varied from 4.93 m$^2$ ha$^{-1}$ to 69.13 m$^2$ ha$^{-1}$. The highest number of trees was found in plot 18 (81 individuals) followed by plot 16 (75 individuals), plot 9 (65 individuals), and plot 14 (61 individuals 61). The lowest number of trees was found in plot 24 (22 individuals) and plot 22 had the most abundant species (five species) as shown in Figure 3. The DBH of individual trees varied between 5 cm and 41.3 cm, with total height ranging from 1.5 m to 9.75 m, with an average of 16.64 ± 8.23 cm

and 5.71 ± 1.90 m. About 50% of the tree diameters and heights were between 10–21.98 cm and 4.267–7.315 m, respectively. Among the six mangrove species generally found at the study site, *A. officinalis* was found to have the maximum DBH (15.80 ± 3.47 cm) and height (5.70 ± 0.85 m). The lowest height and DBH were recorded in *Aegiceras corniculutum* with 12.19 ± 1.83 cm and 3.52 ± 0.61 m respectively. Additionally, Figure 4 described Height–Diameter scatter plot of mangroves at the study site. The functional relationship between height and diameter of a tree was effectively described by a log function. Tree height was positively correlated with diameter of the mangrove stand and the coefficient of determination ($R^2$) was 0.61.

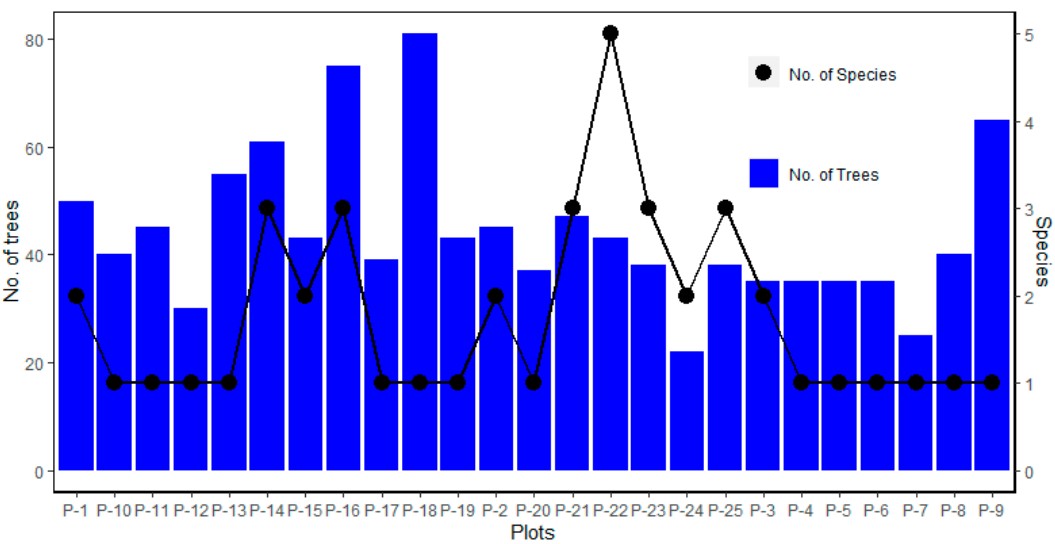

**Figure 3.** Tree abundance and number of species found in different plots in the mangrove stand of LetKhutKon Village.

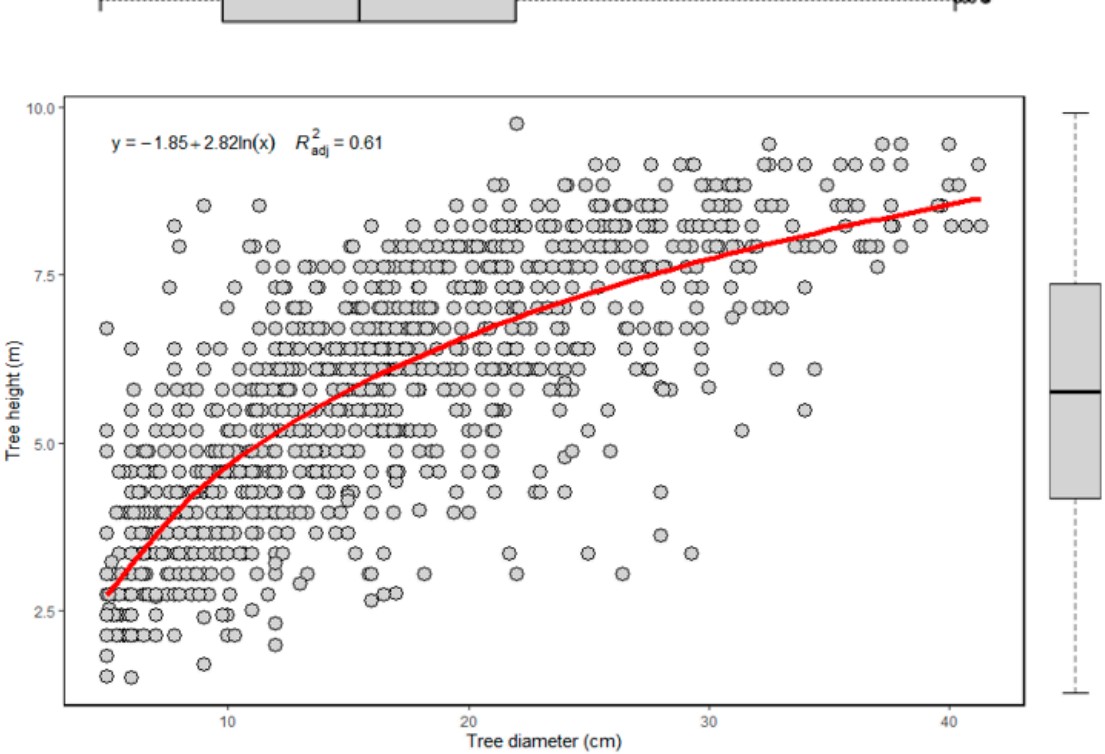

**Figure 4.** Scatter plot of tree height vs diameter in the mangrove stand of LetKhutKon Village.

*3.2. Structural Analysis*

Important value index (IVI) was used to express the dominance and ecological success of any species with a single value; it was determined based on the total contribution of a species to the community by employing its relative density, relative basal area, and relative frequency in a study plot or area [42]. The more the number of individuals found, the higher density values. In the study site, the mangrove species of *A. officinalis* was found to have the highest average stem density ($868 \pm 463$ ha$^{-1}$) and highest relative density of 78.77%, followed by *S. apetala* (13.97%), *S. caseolaris* (4.17%) and *Aegiceras corniculutum* (2.90%). The least mean stem density was recorded by *A. alba* and *Bruguiera sexangula* ($25 \pm 0.00$ ha$^{-1}$). Frequency value of mangrove species is related to the number of plots where mangrove species are found. Here, *A. officinalis* has generally the highest frequency of presence in the study because this species has evenly distributed in each plot (Table 1). The relative frequency of *Aegiceras corniculutum* (11.36%) was higher than that of *S. caseolaris* (9.09%) because *Aegiceras corniculutum* was more evenly distributed than *S. caseolaris* (Table 1). High importance values were owned by the dominant species in a community. Here, *A. officinalis* showed the highest mean basal area ($26.155 \pm 16.940$ m$^2$ ha$^{-1}$), contributing up to 87.66% of the total basal area, and had the highest important value index (IVI) of 218.69%, then followed by 45.53% for *S. apetala*, 15.97% for *S. caseolaris*, 15.06% for *Aegiceras corniculutum*, and 2.37% for *A. alba* and *Bruguiera sexangula* (Table 2). The highest value of importance index of *A. officinalis* explained that *A. officinalis* plays a relatively significant role in maintaining the sustainability of the mangrove ecosystem in the study area.

The genus Avicennia is a pioneer group of dominant plant species and mangrove plants in the genus Avicennia have both economic and ecological values [43]. *A. officinalis* (crypto viviparous) is widely distributed in Bangladesh, Cambodia, India, Indonesia, Malaysia, Myanmar, New Guinea, the Philippines, Sri Lanka, Thailand, Vietnam, and north-eastern Australia [40]. Avicennia species develop pencil-like pneumatophores, while Sonneratia species have thick cone-shaped pneumatophores. *S. apetala* species is also the pioneer species and mainly associated with *A. officinalis* species, [40,44,45]; they are growing on newly formed mudflats near the river mouth and found close to the sea. Therefore, mangrove species of *A. officinalis* and *S. apetala* play a vital role in reducing wave and tidal energy and retaining sediments. In Bangladesh, *Aviecennia officinalis* is planted with *Sonneratia apetala* for the coastal afforestation programme to protect the coastal community against tropical cyclones, storm surges, waves, tides, and saltwater intrusion. In the present study, *A. officinalis* and *S. apetala* have higher density, frequency, stand basal area, and importance values than other species of the mangrove stand; this condition showed that *A. officinalis* and *S. apetala* have high adaptive abilities in the mangrove stand of LetKhutKon Village.

**Table 2.** Tree species found in the mangrove stand of LetKhutKon Village (mean $\pm$ sd).

| Species | Mean Stem Density (No. of Trees ha$^{-1}$) | Mean BA (m$^2$ ha$^{-1}$) | RD (%) | RF (%) | RBA (%) | IVI (%) |
|---|---|---|---|---|---|---|
| *Avicennia officinalis* | $868 \pm 463$ | $26.155 \pm 16.940$ | 78.77 | 52.27 | 87.66 | 218.70 |
| *Sonneratia apetala* | $154 \pm 162$ | $2.633 \pm 4.503$ | 13.97 | 22.73 | 8.83 | 45.53 |
| *Sonneratia caseolaris* | $46 \pm 116$ | $0.807 \pm 1.994$ | 4.17 | 9.09 | 2.71 | 15.97 |
| *Aegiceras corniculutum* | $32 \pm 84$ | $0.238 \pm 0.646$ | 2.90 | 11.36 | 0.80 | 15.06 |
| *Avicennia alba* | $25 \pm 0.0$ | $0.0525 \pm 0.000$ | 0.09 | 2.27 | 0.01 | 2.37 |
| *Bruguiera sexangula* | $25 \pm 0.0$ | $0.050 \pm 0.000$ | 0.09 | 2.27 | 0.01 | 2.37 |

RD is relative density; RF is relative frequency; RBA is relative basal area. The important value is calculated as IVI = RD + RF + RBA and IVI value can add up to a maximum value of 300 [46,47].

Shannon–Wiener index was used to estimate the diversity of species in the study area. The Shannon–Wiener's diversity index (H′) was categorized as low with a value of 0.71 and the Shannon evenness index (SEI) was 0.40. Supporting the results of other studies were the Shannon–Wiener's diversity index value of natural mangrove forest in

the Mahanadi Mangrove Wetland (MMW), East Coast of India was 0.79 ± 0.38 [48], the mangrove of Lauhan village in East Java, Indonesia was 1.51 [24], and mangrove forest in Palawan, the Philippine was 0.99 [3]. Therefore, Shannon–Wiener's diversity index (H′) value of the mangrove community of LetKhutKon Village was very low compared to other natural mangrove forests since the mangrove stand in the study site possessed a low number of mangrove species and was dominated by the few species. In contrast to tropical lowland rainforest, the mangroves have very low diversity as few plants have their special adaptations, which are attributed to their unique stands formation and harsh coastal habitat [8].

### 3.3. Biomass and Carbon Stock of Mangrove

Allometric method is the most widely used method for biomass estimation of the forest because this method provides non-destructive and less time-consuming than other methods [49]. In this study, the parameters of diameter at breast height (DBH) and wood density ($\rho$) were applied to compute mangrove biomass by using the allometric equations of Komiyama et al. [30]. As shown in Table 1, the overall mean biomass of the mangrove stand in LetKhutKon Village was found to be $335.55 \pm 181.41$ Mg ha$^{-1}$ (the average aboveground biomass = $241.37 \pm 132.73$ Mg ha$^{-1}$ and the average belowground biomass = $94.17 \pm 48.73$ Mg ha$^{-1}$) wherein the total biomass produced was 8388.62 Mg ha$^{-1}$. The reported AGB of mangroves in LetKhutKon Village ($241.37 \pm 132.73$ Mg ha$^{-1}$) was comparable to other mangroves, with the values of 255.7 Mg ha$^{-1}$ reported in Lamu, Kenya [50], 246.90 Mg ha$^{-1}$ at Guarás Island located in the state of Para [51] and $80.23 \pm 15.95$ t ha$^{-1}$ at the Kerala state, the southwest corner of India [52]. There were considerable variations in the biomass between different species as shown in Tables 3 and 4. Among the different species, the highest biomass of 7604.607 Mg ha$^{-1}$ was recorded in *A. officinalis* (above and below-ground biomass were 5484.659 Mg ha$^{-1}$ and 2119.947 Mg ha$^{-1}$) and the lowest biomass was in *Avicennia alba*, having 0.333 Mg ha$^{-1}$. The biomasses of remaining species such as *S. apetala*, *S. caseolaris*, *Aegiceras cornoculutum*, and *Bruguiera sexangula* were 597.564 Mg ha$^{-1}$, 135.820 Mg ha$^{-1}$, 49.898 Mg ha$^{-1}$, and 0.397 Mg ha$^{-1}$, respectively.

**Table 3.** Biomass and carbon stock differences among the species in the mangrove stand.

| Species | Biomass (Mg ha$^{-1}$) | | C-Stock (Mg C ha$^{-1}$) | |
|---|---|---|---|---|
| | AGB | BGB | AGC | BGC |
| *Avicennia officinalis* | 5484.659 | 2119.947 | 2577.790 | 826.779 |
| *Sonneratia apetala* | 420.535 | 177.029 | 197.651 | 69.041 |
| *Sonneratia caseolaris* | 94.672 | 41.148 | 44.496 | 16.048 |
| *Aegiceras corniculutum* | 33.951 | 15.947 | 15.957 | 6.219 |
| *Avicennia alba* | 0.213 | 0.120 | 0.100 | 0.047 |
| *Bruguiera sexangula* | 0.256 | 0.141 | 0.120 | 0.055 |

**Table 4.** Mean diameter breast height, biomass, and carbon stock of recorded mangrove species in mangrove stands of LetKhutKon Village (mean ± sd).

| Species | Mean DBH | Biomass (Mg ha$^{-1}$) | | | Vegetation Carbon Stock (Mg C ha$^{-1}$) | | |
|---|---|---|---|---|---|---|---|
| | | AGB | BGB | TB | AGC | BGC | TVC |
| *Avicennia officinalis* | 17.66 ± 8.47 | 6.319 ± 6.888 | 2.442 ± 2.425 | 8.761 ± 9.312 | 2.970 ± 3.237 | 0.953 ± 0.946 | 3.922 ± 4.183 |
| *Sonneratia apetala* | 13.43 ± 6.13 | 2.731 ± 2.885 | 1.150 ± 1.105 | 3.880 ± 3.989 | 1.283 ± 1.356 | 0.448 ± 0.431 | 1.732 ± 1.787 |

**Table 4.** *Cont.*

| Species | Mean DBH | Biomass (Mg ha$^{-1}$) | | | Vegetation Carbon Stock (Mg C ha$^{-1}$) | | |
|---|---|---|---|---|---|---|---|
| | | AGB | BGB | TB | AGC | BGC | TVC |
| *Sonneratia caseolaris* | 13.81 ± 5.79 | 2.058 ± 2.181 | 0.895 ± 0.853 | 2.953 ± 3.034 | 0.967 ± 1.025 | 0.349 ± 0.333 | 1.316 ± 1.358 |
| *Aegiceras corniculutum* | 9.24 ± 3.08 | 1.061 ± 0.836 | 0.498 ± 0.357 | 1.559 ± 1.192 | 0.499 ± 0.393 | 0.194 ± 0.139 | 0.693 ± 0.532 |
| *Avicennia alba* | 5.20 | 0.213 | 0.120 | 0.332 | 0.100 | 0.047 | 0.147 |
| *Bruguiera sexangula* | 5.10 | 0.256 | 0.141 | 0.397 | 0.120 | 0.055 | 0.175 |
| Total | 16.64 ± 8.23 | 5.476 ± 6.438 | 2.136 ± 2.279 | 7.612 ± 8.716 | 2.574 ± 3.026 | 0.833 ± 0.889 | 3.407 ± 3.914 |

The aboveground biomass (AGB) and belowground biomass (BGB) contributed 71.93% and 28.07%, respectively, to the total mangrove biomass. The ratio of BGB to AGB (R:S ratio) ranged from 0.34 to 0.58 and the average ratio of BGB to AGB was 0.44 or 1:2.29. For comparison, the belowground biomass to aboveground biomass (R:S) ratio of mangroves was 0.46 or 1:2.17 in Kerala State, India [52] and 0.38 or 1:2.60 in Samar, the Philippines [3]. Mangrove forests have a higher root: shoot ratio (R: S) (generally R:S ratios between 0.33 or 1:3 and 0.50 or 1:2 [53]) when compared to the upland forests (R:S ratios between 0.22 or 1:4.52 and 0.25 or 1:3.96 [54]). Mangrove species are capable of allocating a high proportion of their total biomass to the belowground components which could be adapted to living in the soft sediments [53]. Figure 5 described the root: shoot (R:S) ratio against tree diameter at breast height (DBH in cm). Trees with DBH 10–21.98 cm had a mean R:S ratio of 0.44 while trees < 10 cm DBH had a mean R:S ratio of 0.52 and trees > 21.98 cm DBH had a value of 0.38. Our findings showed R:S ratio decreased significantly with increasing tree DBH.

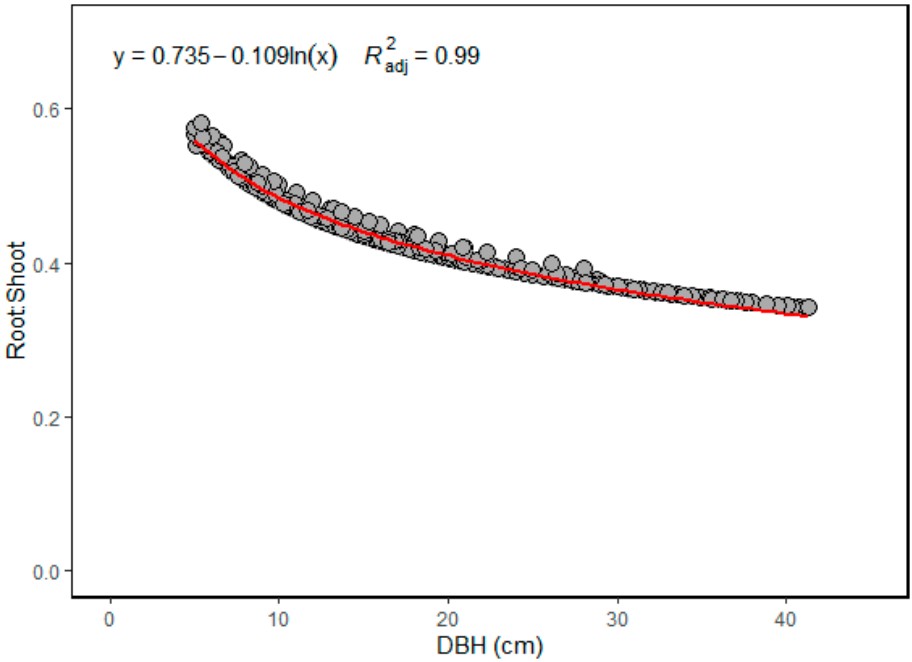

**Figure 5.** Root: Shoot (R:S) ratios against tree diameter at breast height (DBH in cm).

The biomass of the plant is associated with the carbon-storing capacity of the plant [55], so estimating the biomass potential of mangrove vegetation can be used to calculate the carbon stock. The total carbon stock (C-stock) of the mangrove stand in LetKhutKon Village was 3754.304 Mg C ha$^{-1}$ and varied from 17.37 Mg C ha$^{-1}$ to as high as 373.06 Mg C ha$^{-1}$ with a mean value of 150.25 ± 81.35 Mg C ha$^{-1}$. The average C-stock of the mangrove stand

was equivalent to carbon dioxide sequestration of $551.10 \pm 298.64$ Mg $CO_2$ eq (1 ton of carbon = 3.67 tons of carbon dioxide) stored in the biomass. Total aboveground C-stock was 2836.12 Mg ha$^{-1}$ and varied from 12.54 Mg C ha$^{-1}$ to as high as 283.85 Mg C ha$^{-1}$ with a mean value of $113.44 \pm 62.38$ Mg C ha$^{-1}$. The belowground biomass was 2354.33 Mg ha$^{-1}$ with an overall average of $94.17 \pm 48.73$ Mg ha$^{-1}$ and the mean belowground C-stock was $36.73 \pm 19.00$ Mg C ha$^{-1}$ (Table 1). The estimated biomass ($335.54 \pm 181.41$ Mg ha$^{-1}$) and stored carbon ($150.17 \pm 81.37$ Mg C ha$^{-1}$) of mangrove stand in the present study was higher than that of Labuan, Indonesia (168.05 Mg ha$^{-1}$ and 74.7 Mg ha$^{-1}$) [24], Kerala mangrove in Southwest Coast of India (117.1 t ha$^{-1}$ and 139.82 t ha$^{-1}$) [52], mangrove stand in the east coast of India (178.3 t ha$^{-1}$ and 89.1 t ha$^{-1}$) [48], and natural mangrove stands in Bohol Province, Philippines (323.6 t ha$^{-1}$ and 145.6 t ha$^{-1}$) [56]; however, the C-stock estimated in this study was lower than the C-stock obtained in the natural mangrove forest of Bahile, Puerto Princesa City, Palawan (757.7 t ha$^{-1}$ and 356.1 t C ha$^{-1}$) [57], and Thailand (345 t ha$^{-1}$ and 155 t ha$^{-1}$) [58]. The contribution of mangrove species to mean C-stock was in the following order: *A. officinalis > S. apetala > S. caseolaris > Aegiceras corniculutum > Bruguiera sexangula > A. alba* as shown in Tables 3 and 4. Among the established sample plots, the highest biomass and C-stock were attributed in plot-9 with its corresponding maximum stand basal area of 69.125 m$^2$ ha$^{-1}$, whereas the lowest biomass and C-stock occurred in plot-24 with its corresponding minimum stand basal area of 26.692 m$^2$ ha$^{-1}$. Because plot 9 has the highest total DBH among the recorded sample plots; however, the DBH of trees measured in plot 24 was very low, since the trees were newly growing.

### 3.4. The Relationships between Carbon Density and Structural Variables

Pearson's correlation coefficient was used to see the relationship between the variables at a 95% confidence interval. The correlations between the predictor variables and the dependent variable were shown in Table 5. Aboveground carbon stock (AGC) was positively correlated with all structural variables such as mean DBH $(\overline{D})$, mean height $(\overline{H})$, and stand basal area (BA). AGC was positively associated with BA ($R = 0.9921$, $p < 2.2 \times 10^{-16}$), mean DBH ($R = 0.8033$, $p = 3.94 \times 10^{-06}$) and mean height ($R = 0.6838$, $p = 3.21 \times 10^{-04}$); this finding indicated that basal area was a significant predictor of the aboveground carbon-stock of trees in the mangrove stand.

**Table 5.** Pearson's correlation coefficients between aboveground carbon (AGC) density and structural parameters of the stand.

| Structural Variables | Pearson Correlation Coefficient with AGC (Mg C ha$^{-1}$) | *p*-Value |
|---|---|---|
| Mean DBH (cm) | 0.8033 | $3.94 \times 10^{-06}$ |
| Mean H (m) | 0.6838 | $3.21 \times 10^{-04}$ |
| BA (m$^2$/ha) | 0.9921 | $<2.2 \times 10^{-16}$ |

### 3.5. Influence of Structural Variables on Aboveground Carbon-Stock

Linear regression analysis was performed to describe the relationship between stand-level carbon stocks (Mg C ha$^{-1}$) as the dependent variable and stand structural parameters such as mean DBH, mean H and basal area as independent variables. All models were named and described in Table 6. As specified in the Table, carbon stock was significantly correlated with structural variables. Through the linear regression analysis, we found that carbon stored in the tree biomass was influenced by forest structural characteristics.

**Table 6.** Linear regression analysis result of stand structural variables and aboveground carbon stock (AGC).

| Model | Adj. $R^2$ (%) | RMSE | AIC | BIC | bptest | CF | *p*-Value |
|---|---|---|---|---|---|---|---|
| Model 1 | 67.92 | 0.267 | 10.570 | 13.977 | 0.521 | 1.0398 | $8.14 \times 10^{-07}$ |
| Model 2 | 46.25 | 0.346 | 22.440 | 25.847 | 0.382 | 1.0677 | 0.000214 |
| Model 3 | 97.21 | 0.079 | −45.586 | −42.180 | 0.230 | 1.0034 | $<2.2 \times 10^{-16}$ |
| Model 4 | 97.28 | 0.076 | −45.350 | −40.808 | 0.187 | 1.0033 | $<2.2 \times 10^{-16}$ |

Note: Model 1: one-variable (mean DBH (cm)), Model 2: one-variable (mean Height, (m)), Model 3: one-variable (basal area, (m$^2$)), Model 4: two-variable (BA, $\overline{\text{H}}$ (m$^2$, m)). The statistics represent the coefficient of determination (R$^2$-adj), Root Mean Square Error (RMSE), Akaike information criterion (AIC), Bayesian Information Criterion (BIC), bptest, Correction factor (CF) and *p*-value.

Model (1) analyzed the relationship between aboveground carbon stock-AGC (Mg C ha$^{-1}$) and mean diameter at breast height. The model had R$^2$-adj = 0.6792, AIC = 10.570, BIC = 13.977, RMSE = 0.267 and $p = 8.14 \times 10^{-07}$. As mean DBH increased by 1 cm, on an average aboveground carbon stock increased by 2.1231 Mg ha$^{-1}$ keeping all things constant; it showed AGC had a direct correlation with DBH, therefore it could be assumed that DBH was a reliable dendrometric variable for aboveground carbon stock estimation [59–61]. The mangrove tree biomass model, which was determined from DBH, only had a practical advantage because most of the inventories included DBH measurements. Furthermore, it was easy to measure accurately in the field.

$$\ln(\text{AGC}) = -1.3088 + 2.1231 \ \ln(\overline{\text{D}}) \tag{21}$$

The relationship between aboveground carbon stock-AGC (Mg ha$^{-1}$) and mean height (m) was assessed in model 2; this model had a coefficient of determination of 0.4625 and the parameters were statistically significant ($p = 0.0002142$). Although AGC had a significant positive relationship with Mean H, it showed a lower R$^2$-adj (46.25%), higher AIC (22.440), and higher BIC (25.847) when compared with the relationship between AGC and mean DBH; thus, mean height (m) as an individual independent variable was deniable as one of the important predictors for the estimation of AGC.

$$\ln(\text{AGC}) = 1.3728 + 1.8867 \ \ln(\overline{\text{H}}) \tag{22}$$

The result of the linear regression analysis of the model (3) revealed that aboveground carbon stock density (Mg ha$^{-1}$) had a significant, positive relationship with stand basal area (m$^2$ ha$^{-1}$) with a coefficient of determination of 0.9834; parameters were statistically significant ($p < 2.2 \times 10^{-16}$). The result was statistically interpreted, as the stand basal area increased by 1 m$^2$ ha$^{-1}$, and on an average above-ground carbon stock increased by 1.21227 Mg ha$^{-1}$ keeping all things constant. The strong relationship between stand basal area and aboveground carbon stock is because both variables have been associated with the diameter of a tree trunk; it means if the size of the tree trunk increases, the stand basal area increases because tree basal area is the cross-sectional area of a tree trunk measured at the breast height over bark, and as a consequence, the aboveground biomass and carbon stock also increase.

$$\ln(\text{AGC}) = 0.58368 + 1.21227 \ln(\text{BA}) \tag{23}$$

Across all structural variables, stand-level carbon stock showed the highest relationship (R$^2$ adj = 97.21, $p < 2.2 \times 10^{-16}$) with the stand basal area. When the stand basal area and tree height were used as compound variables in the model, it explained 97.28% of carbon variation. Model (4) has appeared the best fit model showing fitting statistics (R$^2$ = 97.28, AIC = −5.350, BIC = −40.808, RMSE = 0.076, $p$-value $< 2.2 \times 10^{-16}$) and very close to model 3. Despite that fact, both models (3 and 4) were still able to explain carbon storage very well (Adj. R$^2$ > 90%).

$$\ln(\text{AGC}) = 0.46971 + 1.16254 \ \ln(\text{BA}) + 0.16235 \ \ln(\overline{\text{H}}) \tag{24}$$

For a better analysis of residual distribution, we used the respective statistical test for checking the models. We ran the Durbin–Watson test to detect the autocorrelation in the residuals. The test statistics were for model 3: DW = 1.7032, *p*-value = 0.2052 and for model 4: DW = 1.5571, *p*-value = 0.1014. For both models, the presence of autocorrelation was not significant as the *p*-value > 0.05 and the value of DW ~ 2. We further ran the Shapiro–Wilk test for normality and both models were normality distributed (*p*-value = 0.8458 and 0.7164 for model 3 and model 4 respectively). Additionally, we performed Breusch–Pagan test (bptest) to determine whether heteroscedasticity was present in the regression model. The test statistics concluded there may not be heteroscedasticity as the *p*-value > 0.05; thus, the best fit equations for estimating stand-level carbon stock were Model 3: ln(AGC) = 0.58368 + 1.21227 ln(BA), one-variable model using stand basal area (m$^2$ ha$^{-1}$) as predicted variable, and Model 4: In(AGC) = 0.46971 + 1.16254 ln(BA) + 0.16235 ln ($\overline{\text{H}}$), two-variable model using stand basal area and mean height (m$^2$, m).

## 4. Conclusions

By applying the non-destructive methodology, biomass and carbon stock of the mangrove stand of Kanhlyashay natural mangrove forest were estimated. A low diversity index value (H′ = 0.71) was observed in the natural mangrove stand that was dominated by the species of *Aviecennia officinalis* (IVI = 218.69%) from Acanthaceae family comprised 78.77% of the total tree count. Therefore, *A. officinalis* has high adaptive abilities in the mangrove stand of LetKhutKon Village. Pioneer mangrove species such as *Avicennia officinalis* and *Sonneratia caseolaris* have a good survival rate on the mudflats, and they are suitable mangrove species for mangrove afforestation on unoccupied mudflats because of their tolerance to increased salinity. The total biomass and carbon stock in the natural mangrove forest were 335.55 ± 181.41 Mg ha$^{-1}$ and 150.25 ± 81.37 Mg C ha$^{-1}$, where the above and below-ground carbon stock contributed 71.93% and 28.07%, respectively. Stand-level allometric equations in the estimation of aboveground carbon stock were implicated. The finding revealed that the one-variable model of the stand basal area and the two-variables model (basal area + mean height) were suitable based on fitting statistics and certain statistical tests for high-precision estimates of stand-level carbon stock of Mangrove stand in the study site. Our observation highlights the natural mangrove forest in the study site has the potential to store and sequester a significant amount of carbon. Because natural mangrove forest in the study site is a young age stand and is dominated by the fast-growing pioneer species. At a young age stand, the rate of carbon sequestration is high. In addition to the stand age, the rate of carbon uptake of the forest ecosystem depends on forest management. Therefore, forest management activities are necessary to maintain forest carbon sequestration capacity.

**Author Contributions:** Conceptualization, W.N.A. and X.T.; data collection, methodology, validation, formal analysis, data curation, writing—original draft preparation, W.N.A.; writing—review and editing, W.N.A., X.T. and A.W.T.; supervision, X.T. All authors contributed critically to the drafts and gave final approval for publication. All authors have read and agreed to the published version of the manuscript.

**Funding:** Article Processing Charge(APC) was funded by National Natural Science Foundation of China (31872703).

**Acknowledgments:** The authors gratefully acknowledge APFNet Scholarship funding the first author's research at Beijing Forestry University. The authors thank to everyone who helped with the field survey and the anonymous reviewers for their valuable comments.

**Conflicts of Interest:** The authors declare no conflict of interest.

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
