# Peer review of "Species Diversity, Biomass and Carbon Stock Assessment of Kanhlyashay Natural Mangrove Forest"

_forests, doi:10.3390/f13071013_

Round 1
Reviewer 1 Report
Dear Authors,
Your manuscript is a valuable contribution with high scientific output.
I propose you to do some transformation of the text, to give explanations and make a few corrections. The details are presented herewith below.
Lines 31-33
Climate change mitigation may be achieved not only by reducing the carbon emission levels but also by maintaining the mangrove ecosystem services as carbon sinks and sequestration.
Important statement founded a possibility of climate correction. This is especially relevant against a background of endless standard claims about the danger of climate change. I propose you to substantiate your point of view in details by showing the ways to reinforce a biological buffer of the climate system. In my opinion, a broader interpretation of your results is quite justified, since a biological buffer of the climate system can be strengthened in various parts of the land. This approach is our Biogeosystem Technique methodology you can acknowledge in our publications: https://pubs.acs.org/doi/10.1021/acsomega.0c02014
https://doi.org/10.1016/j.envres.2020.110605
https://doi.org/10.1021/scimeetings.0c07087
Line 25
1-ha
This unit of measure is not a best choice.
Lines 301-303
Therefore, Shannon-Wiener’s diversity index (H ̍) value of the mangrove community of LetKhutKon Village was very low compared to other natural mangrove forests...
Please explain your statement in more details since a low diversity is usually considered a disadvantage of the ecosystem.
Please do not use two different notations for the denominator of a fraction: Mg ha-1; m2/ha.
In Conclusions section, you briefly repeated a content of above sections of the manuscript, emphasizing an importance of statistical models. It is too narrowed approach. I recommend you to answer your statement on climate regulation you made in the Introduction section.
A vital circumstance of forest system is a fact that its carbon sequestration is limited. In the young forest stand, a rate of sequestration is high. However, a forest stand at a climax stage becomes a source of carbon emission to the atmosphere. It should be great if you would discuss this circumstance in your manuscript.
Reviewer 2 Report
The structure of the article is generally typical for this type of work. However, it requires a few corrections.
1. The title of the article should be shortened, ending with the word “forest”. Further information related to the location should be included in the methodological part.
2. The abstract is too detailed and should be redrafted. Moreover, no citations should be given in this section.
3. Introduction should be extended to include matters related to carbon sequestration. Citations should generally be placed at the end of sentences.
4. Please justify choosing only twenty-five sampling plots covered 0.5% of the total area. Is it a number that has been calculated on the basis of the variability of the stand features? In my opinion, there should be more of these areas.
5. I suggest separating Discussion from Results and create the next chapter where Authors compare their results to similar ones done by other researchers.
6. Table 1 containing the measurement data should be transferred to the appendices. However, on the basis of these data, an appropriate synthesis should be made.
7. Fig. 4. I suggest replacing the figure with a table, because some of the results, close to 0, are almost invisible.
Detailed comments:
1. Too large font of the legend in Fig. 1.
2. Line 142. Please use m2 / ha instead m2 / ha.
3. Please standardize the form of citations in line with the journal's requirements.
4. Fig. 3. Please also provide the percentages, as the species marked with gray and green colors are almost invisible.
Round 2
Reviewer 2 Report
The article can be published in present form. The Autors have taken my suggestions into account. Although, personally, I prefer to separate the results from discussion.